# Nuclear Functions of KaeA, a Subunit of the KEOPS Complex in *Aspergillus nidulans*

**DOI:** 10.3390/ijms231911138

**Published:** 2022-09-22

**Authors:** Joanna Gawlik, Michal Koper, Albert Bogdanowicz, Piotr Weglenski, Agnieszka Dzikowska

**Affiliations:** 1Institute of Genetics and Biotechnology, Faculty of Biology, University of Warsaw, Pawińskiego 5A, 02-106 Warsaw, Poland; 2Centre of New Technologies, University of Warsaw, Żwirki i Wigury 93, 02-089 Warsaw, Poland; 3College of Inter-Faculty Individual Studies in Mathematics and Natural Sciences, University of Warsaw, Banacha 2c, 02-097 Warsaw, Poland; 4Institute of Biochemistry and Biophysics, Polish Academy of Sciences, Pawińskiego 5A, 02-106 Warsaw, Poland

**Keywords:** KEOPS/EKC complex, transcription regulation, RcoA^Tup1^ co-repressor

## Abstract

Kae1 is a subunit of the highly evolutionarily conserved KEOPS/EKC complex, which is involved in universal (t6A_37_) tRNA modification. Several reports have discussed the participation of this complex in transcription regulation in yeast and human cells, including our previous observations of KaeA, an *Aspergillus nidulans* homologue of Kae1p. The aim of this project was to confirm the role of KaeA in transcription, employing high-throughput transcriptomic (RNA-Seq and ChIP-Seq) and proteomic (LC-MS) analysis. We confirmed that KaeA is a subunit of the KEOPS complex in *A. nidulans.* An analysis of *kaeA19* and *kaeA25* mutants showed that, although the (t6A_37_) tRNA modification is unaffected in both mutants, they reveal significantly altered transcriptomes compared to the wild type. The finding that KaeA is localized in chromatin and identifying its protein partners allows us to postulate an additional nuclear function for the protein. Our data shed light on the universal bi-functional role of this factor and proves that the activity of this protein is not limited to tRNA modification in cytoplasm, but also affects the transcriptional activity of a number of nuclear genes. Data are available via the NCBI’s GEO database under identifiers GSE206830 (RNA-Seq) and GSE206874 (ChIP-Seq), and via ProteomeXchange with identifier PXD034554 (proteomic).

## 1. Introduction

The Kae1/KaeA protein is a subunit of the highly evolutionarily conserved kinase, endopeptidase, and other proteins of small size/endopeptidase-like and kinase associated to transcribed chromatin (KEOPS/EKC) complex, and has been identified in all previously sequenced genomes of *Archea*, *Bacteria*, and *Eucaryota*, except for the most reduced genomes [1]. The KEOPS complex consists of five subunits. In addition to Kae1, it comprises three auxiliary proteins highly conserved only in *Archea* and *Eucaryota*: Bud32, an atypical serine-threonine protein kinase [2], Cgi121, and Pcc1p. The fifth subunit is an intrinsically disordered protein, Gon7, which is less evolutionarily conserved and specific to *Eucaryota* (for review, see [3]). In *Eucaryota*, a mitochondrial paralogue of Kae1 named Qri7p was also found [4].

The complex catalyzes the universal and essential modification of tRNA, converting adenosine A_37_ immediately 3′ of anticodon to N6-threonylcarbamoyl adenosine (t6A_37_) [5,6,7,8,9]. This type of modification is found in tRNA molecules with anticodons decoding ANN codons, and this is one of the few tRNA modifications present in all organisms [10]. t6A_37_ stabilizes the A-U codon-anticodon interaction, prevents frameshifting during translation, and enables proper translation initiation at the AUG start codon, as it is also related to the initiator tRNA_i_^Met^ [11]. The modification also requires Sua5, which is highly conserved in all organisms but is not a part of the KEOPS complex [12]. Sua5 catalyzes the synthesis of threonylcarbamoyl adenylate intermediate, which is used by Kae1 for tRNA modification [8]. A crystal structure-based model of the yeast KEOPS complex has also been proposed [9].

The primary function of the KEOPS complex appears to be t6A_37_ modification. However, several mutations in genes encoding KEOPS subunits have been described, the phenotype of which suggests that the complex subunits may have other functions. The group of proteins interacting with only two KEOPS subunits, PRPK (human homologue of Bud32) and TPRKB (human homologue of Cgi121), was shown to be enriched in proteins implicated in translation, suggesting that the minor PRPK-TPRKB complex may be involved in translation-related processes. Proteomic analysis showed that the PRPK-TPRKB complex interacts with the m^1^A_58_ tRNA methyltransferase complex, which methylates adenosine A_58_ on the initiator tRNA_i_^Met^, suggesting that the PRPK-TPRKB complex may be involved in both t6A_37_ and m^1^A_58_ tRNA modification [13].

The KEOPS complex was implicated in telomere elongation, uncapping, and recombination [14,15]. The complex was proposed to promote an “open” telomere conformation, allowing access for both telomerase and exonuclease. KEOPS was also shown to positively regulate telomere length independent of its tRNA modification function [16]. The interaction of Bud32p with Grx4p glutaredoxin suggested additional functions of this kinase in yeast [17]. Chromatin immunoprecipitation experiments performed with yeast Pcc1p [18] and human OSGEP and LAGE3 [19], homologues of Kae1 and Pcc1, showed that these subunits are localized in chromatin, and it was suggested that they participate in chromatin structure/gene regulation.

In the model filamentous fungus *Aspergillus nidulans*, mutations in the *kaeA* gene (previously called *suDpro*) were identified as suppressors of proline auxotrophic (*pro^−^*) mutations [20]. Two characterized *kaeA* mutations (previously called *suD19pro* and *suD25pro*) result in upregulation of *agaA* and *otaA* genes, coding for arginine catabolic enzymes: arginase and ornithine aminotransferase, respectively [21,22]. Both of these mutations are short deletions in the *kaeA* coding region, which do not change the reading frame. *kaeA19* is the deletion of 12 base pairs, resulting in a lack of four amino acids (STPQ) at the N terminus of the protein, while *kaeA25* (deletion of nine base pairs) results in the replacement of four amino acids (KTGF) by one (N) at the C terminus. These mutations do not change the sequence of the conserved regions of the protein. A model of the KaeA structure shows that amino acids from both deleted regions are localized on the protein surface, and it has been proposed that these deletions influence the interaction of KaeA with KEOPS subunits or other proteins [22].

Preliminary transcriptomic analysis showed that *kaeA25* mutation changed the level of expression of several genes involved in various domains of cellular metabolism, such as amino acid/siderophore and carbon/energy metabolism. Our previous results supported the hypothesis that in addition to its conserved role in tRNA modification, KaeA might be involved in transcription regulation. We postulated that the expression of some genes, e.g., arginine catabolism genes, might be directly regulated at the transcription level by KaeA, although, most probably, several observed pleiotropic effects of *kaeA19/kaeA25* mutations were indirect [22].

In this paper, in order to define the role of KaeA in transcription regulation, we undertook in-depth transcriptomic analysis, comparing the transcriptomes of wild-type and *kaeA19*/*kaeA25* mutant strains, proteomic analysis to identify proteins interacting with KaeA, and ChIP-Seq analysis to identify the chromatin regions KaeA interacts with. We confirmed that KaeA is a subunit of the KEOPS complex in *A. nidulans*. Although the level of t6A_37_ modification is not affected in *kaeA19* and *kaeA25* mutants, RNA-Seq analysis confirmed that, compared to the wild type, the expression level of several hundred genes is changed in the mutants. We found corroborating evidence showing that, similar to what was shown in yeast and human cells, KaeA is localized in chromatin. Finally, protein partners of KaeA were identified, allowing us to postulate an additional nuclear function of the protein.

## 2. Results

### 2.1. kaeA19 and kaeA25 Mutations Do Not Influence the Modification of A_37_ in tRNA

*A. nidulans kaeA19* and *kaeA25* mutants grow much more slowly than the wild type [22], but the effect is not so severe, as in *S. cerevisiae,* where the haploid *kae1* deletion strain is viable but extremely slow growing and can be stored only in diploid *kae1*Δ*/kae1^+^* [7]. Therefore, the influence of both *kaeA19* and *kaeA25* mutations on t6A_37_ modification was determined by using a modified primer extension experiment, according to the procedure described by Srinivasan et al., for *S. cerevisiae* [7].

Both *kaeA* mutants and respective control isogenic *kaeA^+^* strains were used for analysis. *S. cerevisiae kae1*Δ and the wild-type strain were used as positive controls for the experiment. A fraction of small RNAs was purified from *A. nidulans* and *S. cerevisiae* strains and subjected to primer extension analysis using primers specific for tRNA^Ile^ containing the t6A_37_ modification. Primer specific for tRNA^Val^ not modified in this way was additionally used for *A. nidulans* samples as a negative control. 

As expected, in yeast, the wild-type and *kae1*Δ strains had different patterns for tRNA^Ile^ at the position corresponding to t6A_37_ modification. However, for both *A. nidulans kaeA* mutants, the differences were rather unnoticeable (Figure 1). This indicates that both *kaeA19* and *kaeA25* mutations have no significant effect on t6A_37_ modification of tRNA^Ile^.

### 2.2. kaeA19 and kaeA25 Mutations Affect Expression of Many Genes

As mentioned above, preliminary transcriptomic analysis conducted for only one *kaeA* mutant (*kaeA25*) and the wild-type strain showed significant changes in the expression of several genes in the mutant [22]. These results were obtained using pyrosequencing, which delivers a much smaller number of reads than the Illumina technology used nowadays. Therefore, to achieve truly quantitative transcriptomic data, we repeated RNA-Seq analysis of both *kaeA19* and *kaeA25* mutants using the Illumina NextSeq platform and performed independent sequencing of cDNA libraries for each of the three independent biological experiments, in contrast to the preliminary analysis, where RNA-Seq libraries from three independent biological experiments were pooled before the sequencing step.

To identify differentially expressed genes in *kaeA19* and *kaeA25* mutants, transcriptomes of *kaeA* mutants and control isogenic strains grown on minimal medium were compared. Significant changes in gene expression were noted in both mutants, with 341 upregulated and 134 downregulated genes in *kaeA19*, and 92 and 63, respectively, in *kaeA25* compared to the wild type (Appendix A). As expected, arginine catabolism genes *agaA* and *otaA* were identified among the upregulated genes in both mutants, confirming the reliability of the transcriptomic analysis. Functional enrichment analysis was performed for all genes differentially expressed in *kaeA19* and *kaeA25* mutants, which indicated an overrepresentation of genes involved in nitrogen metabolism, especially amino acid metabolism, and carbon/energy metabolism (Appendix A).

The observed changes in the expression levels of so many genes do not seem to be due to the direct influence of KaeA on their transcription. It seems more likely that KaeA affects, among other aspects, the level of transcription of genes encoding regulatory proteins, specific or wide-domain, which in turn leads to changes in the level of transcription of their downstream target genes. To additionally confirm this hypothesis and the results of RNA-Seq, RT-qPCR analysis of selected differentially expressed genes encoding regulatory proteins or potential regulatory proteins was performed. The analysis was conducted for *rosA, nosA*, and *jlbA*, coding for characterized transcriptional regulators; *nmrA*, coding for co-repressor of general nitrogen transcriptional regulator AreA; and AN2366, coding for protease participating in NmrA cleavage [23] and several uncharacterized genes that, according to FungiDB and SGD, potentially code for regulatory proteins, i.e., AN0948, AN1298, AN6076, and AN10854 (see Table 1 for specific gene function).

RT-qPCR analysis was performed using total RNA from mycelia of the *kaeA19* and *kaeA25* mutants and the respective isogenic wild-type strains. *kaeA19* mutation resulted in increased expression of *rosA*,*nosA*, *jlbA*, AN0948, AN6076, and AN2366 genes, while the expression of *nmrA* and AN1298 decreased. In *kaeA25* mutant, similar changes of expression were observed for *nosA*, AN0948, and AN6076, and additionally, in this mutant a lower transcript level of AN10854 was detected (Figure 2). These results show that KaeA affects the level of transcription of several genes encoding wide-domain regulatory proteins, indirectly influencing the expression of the genes regulated by them.

### 2.3. In Addition to Subunits of KEOPS Complex, KaeA Also Interacts with Other Proteins

An important point in assessing whether KaeA is involved in the regulation of transcription was identifying its potential protein partners. To identify proteins interacting with KaeA, the strain expressing KaeA-TEV-GFP fusion (6) was obtained (see Section 4, Section 4.4). In such a fusion, the TEV protease recognition site is inserted between the KaeA and GFP sequences. The use of GFP-Trap chromatography resin allowed the isolation of proteins interacting with KaeA; however, proteins non-specifically interacting with GFP could also be isolated. Cleavage with TEV protease instead of the standard elution step allowed separation of KaeA from GFP- and GFP-interacting proteins. Such an approach significantly increases the chance of obtaining proteins that specifically interact with KaeA.

The strain expressing KaeA-TEV-GFP fusion and the control isogenic strain (1) were grown on minimal medium. Four independent biological experiments for each strain were performed. Two were conducted with the addition of DNase during the purification procedure to identify protein–protein interactions occurring in the presence of chromatin. Proteomes of the strain expressing KaeA-TEV-GFP fusion protein and the control isogenic strain were compared. A total of 89 proteins were selected that were not present in the control strain and were present at least twice in the strain expressing the KaeA–TEV–GFP fusion, including at least once with Mascot score > 70 (Appendix A).

As expected, we identified KaeA and the other four subunits of the KEOPS complex: PipA (Bud32p homologue); AN2845 (Pcc1p homologue), named PccA; AN11910 (Cgi121p homologue), named Cgi121; and AN11901 (Gon7p homologue), named GonG. This confirmed the reliability of the proteomic analysis. Among potential protein partners of KaeA, we identified protein kinases (CmkA, PkaA/PkaR, PkcB), putative protein phosphatase, and several proteins involved in translation, such as translation initiation factors, prolyl_tRNA-synthetase and the yeast GCN20 homologue, mRNA of which was also identified by transcriptomic analysis. Moreover, a few proteins involved in transcriptional regulation were also identified, including two subunits of general transcriptional co-repressor RcoA and SsnF, and AN2181, a homologue of the yeast Toa2p–TFIIA small subunit (Table 2).

### 2.4. KaeA Interacts with Chromatin

The direct participation of KaeA in transcription regulation requires its interaction with chromatin. To identify chromatin regions which KaeA interacts with, the strain expressing KaeA-HA fusion (7) was constructed (see Section 4, Section 4.4) and chromatin immunoprecipitation coupled with next-generation sequencing (ChIP-Seq) was performed.

Usually, ChIP-Seq is carried out for proteins for which at least one DNA target sequence is known. Before proceeding to the next-generation sequencing step, such a sequence is used in a control qPCR reaction to confirm that the ChIP experiment was conducted correctly. The difficulty in conducting this experiment with KaeA is that it is not a DNA-binding protein, and its interaction with chromatin most probably occurs via other proteins that directly bind to DNA. To validate the ChIP reaction, it was carried out in parallel for two strains: (7), expressing KaeA-HA, and (9), expressing AreA-HA. AreA is a general nitrogen regulator in *A. nidulans,* a well-characterized GATA transcription factor for which many target sequences are known (reviewed in [24]). Prior Western analysis using anti-HA antibodies confirmed that both fusions are expressed (Appendix A). The control qPCR reaction was conducted for immunoprecipitated (Ab) and non-immunoprecipitated (No Ab) samples from strain expressing AreA-HA fusion, using primers specific for bi-directional *niiA–niaD* promoter, as described by Berger et al., [25]. Significant enrichment (Ab/No Ab = 25.7) of the *niiA–niaD* promoter was observed in immunoprecipitated compared with non-immunoprecipitated samples, confirming that the ChIP procedure was correctly carried out. Then, processed in parallel, both types of samples from strain expressing KaeA-HA fusion, immuno-precipitated and non-immuno-precipitated (input control) were subjected to the next-generation sequencing step of ChIP-Seq analysis. Input DNA before immunoprecipitation was used as a control to assess the enrichment of specific DNA regions after immunoprecipitation.

Using the selection criteria described in Section 4, 104 chromatin regions interacting with KaeA were identified, 78 of which were localized within the identified genes (Appendix A). Functional enrichment analysis was carried out, showing that genes involved in nitrogen, sulfur, and lipid metabolism were overrepresented (Appendix A). Inter alia, KaeA interaction with chromatin was demonstrated in the promoter and 5′ UTR regions of genes coding for histones, enzymes involved in nitrogen and carbon metabolism, and transcription factors CpcA and HapX (Table 3).

## 3. Discussion

The function of the exceptionally evolutionarily conserved KEOPS complex, and particularly Kae1, in the t6A37 modification of tRNA is well known and documented [3,9]. In the previous paper, we characterized Kae1 encoding gene, named *kaeA*, in the model filamentous fungus *A. nidulans* [22]. In fact, with the use of classical genetics methods, the gene was identified over 50 years ago as a suppressor of proline auxotrophic (*pro^−^*) mutations [20] and was proposed to be involved in regulation of arginine catabolism genes [21]. In this paper, we present result confirming that KaeA is a subunit of the KEOPS complex in *A. nidulans*. Using a GFP-Trap^®^A resin and the strain expressing KaeA-GFP protein fusion, proteins interacting with KaeA were purified, and subsequently identified using LC-MS analysis. As expected, the remaining subunits of the KEOPS complex were identified among the interacting proteins: PipA (Bud32p homologue); AN2845 (Pcc1p homologue), named PccA; AN11910 (Cgi121p homologue), named Cgi121; and AN11901 (Gon7p homologue), named GonG (Table 2). This result confirms that, in *A. nidulans*, as in other organisms, the primary function of KaeA is the modification of tRNA.

However, *kaeA19* and *kaeA25* mutations result in upregulation of arginine catabolism genes, the transcription level of which is high in both mutants, despite the lack of an arginine inducer in the medium [22]. This was the basis for the hypothesis that KaeA is directly involved in the regulation of these genes, although some indirect effects related to its translational function could not be ruled out. To verify whether the phenotype of *kaeA19* and *kaeA25* mutants might be related to the defective function of the KEOPS complex, a primer extension experiment was performed to detect t6A_37_ modification in tRNA^Ile^, according to the method described for yeast [7]. *A. nidulans kaeA* mutants and *S. cerevisiae kae1*Δ strain were analyzed and, as expected, the wild-type yeast and *kae1*Δ strains differed in their pattern at a position corresponding to t6A_37_ modification (Figure 1). However, no significant difference was observed for *A. nidulans kaeA* mutants, suggesting that they probably have no effect on the level of t6A_37_ modification. The t6A_37_ modification was also studied in *Drosophila melanogaster* [26]. In this organism, several point mutations in the *kae1* gene result in an extended larval period prior to lethality. Similarly, a substantial level of t6A_37_-modified tRNAs was detected in the *Drosophila kae1* mutants, although it should be noted that none of these mutations are located in the same region of the protein as *kaeA19* and *kaeA25.*

Since the beginning of research on the KEOPS complex, its link with the regulation of transcription has been suggested. Genetic interaction analysis showed that yeast Pcc1p interacts with RNA polymerase II and general transcription factor co-activators. Chromatin immunoprecipitation demonstrated that this KEOPS subunit, similarly to Kae1p, is associated with transcribed genes and is required for efficient recruitment of TBP, SAGA and Mediator complexes to α factor and galactose induced promoters [18]. Chromatin localization was also shown for OSGEP and LAGE3, the human homologues of Kae1p and Pcc1p [19]. It was suggested that the KEOPS complex might be responsible for the regulation of chromatin structure, possibly due to the ATP-dependent remodeling activity of Kae1p [18].

Our previous results, including preliminary transcriptomic analysis, also suggested some transcriptional functions of KaeA; however, the possibility that the observed effects resulted from defects in KEOPS function could not be ruled out. Here, we present the results of repeated transcriptomic analyses of *kaeA19* and *kaeA25* mutants using the Illumina NextSeq platform. We obtained 100-fold coverage of the protein coding sequences, which is ten times more than in the first analysis. New transcriptomic data confirm that, although t6A_37_ modification in *kaeA19* and *kaeA25* mutants is not affected, their transcriptomes significantly differ from the wild-type transcriptome.

It is rather unlikely that the changes in expression level of several hundred genes observed in *kaeA19* and *kaeA25* mutants resulted from the direct influence of KaeA on their transcription. Among differentially expressed genes, several coding for regulatory proteins were selected, and RT-qPCR analysis additionally confirmed that *kaeA19/kaeA25* mutations affected their transcription level (Table 1). In this way, KaeA can indirectly influence the expression of many more genes regulated by these regulatory proteins, including arginine catabolism genes.

Many genes whose transcription level is altered in *kaeA19* and/or *kaeA25* mutants are involved in carbon/energy and nitrogen metabolism, especially amino acid metabolism (Appendix A). Downregulation of *nmrA,* correlated with upregulation of AN2366, coding for protease involved in specific inactivating cleavage of NmrA [23], explains the indirect effect of *kaeA19* mutation on many nitrogen metabolism genes, as NmrA is a co-repressor of the wide-domain nitrogen transcriptional regulator AreA [27,28,29]. An indirect effect of *kaeA19* on amino acid biosynthesis genes is likely due to upregulation of *jlbA,* encoding the bZIP transcription factor involved in their regulation [30], and AN9048, orthologue of yeast *GCN20,* coding for a positive regulator of Gcn2p protein kinase involved in cross-pathway regulatory network of amino acid biosynthesis [31]. Strong downregulation of *AN10854,* an orthologue of yeast *SNF4*, encoding the activating subunit of the Snf1p protein kinase complex involved mainly in glucose, but also nitrogen sensing [32], might explain the indirect effect of *kaeA25* on the expression of many carbon/nitrogen metabolism genes.

The indirect influence of KaeA on the expression of many other genes may also result from its influence on the expression of genes encoding proteins involved in chromatin structure regulation, i.e., *rosA* and *nosA*, orthologues of yeast *UME6,* encoding a subunit of the Rpd3L histone deacetylase complex involved in chromatin remodeling and transcriptional repression [33], and AN6076, an orthologue of yeast *SWR1,* encoding a catalytic subunit of the nucleosome remodeling complex SWR1 [34].

Interestingly, in the *kaeA19* downregulation of AN1298, an orthologue of yeast *RTG3*, coding for RTG and TOR-controlled transcription activator [35,36], was also observed. This may explain the earlier described pleiotropic effects of *kaeA19* mutation related to mitochondrial defects, such as impaired growth on glycerol, altered mitochondrial structure, and higher sensitivity to ethidium bromide [22].

Important evidence that KaeA is involved in the regulation of transcription was the identification of chromatin regions which KaeA interacts with, using the strain expressing KaeA-HA fusion and chromatin immunoprecipitation coupled with next-generation sequencing (ChIP-Seq). Similarly to what was shown in yeast [18] and human [19] cells, we demonstrated that KaeA is localized in chromatin. Among the chromatin regions interacting with KaeA, 78 were localized within the identified genes, in most cases within a sequence defined as 5′ UTR (Table 3 and Appendix A), with overrepresentation of genes involved in nitrogen, sulfur, and lipid metabolism. This includes genes encoding key nitrogen metabolism enzymes (glutamine synthetase, NADP^(+)^-dependent glutamate dehydrogenase, and high-capacity ammonium permease) as well as glyceraldehyde-3-phosphate dehydrogenase (GAPDH), involved in glycolysis and gluconeogenesis. KaeA was also found in a chromatin region comprising the *cpcA* gene, a yeast *GCN4* orthologue, encoding central transcription factor of the cross-pathway regulatory network of amino acid biosynthesis [37]. Interestingly, among chromatin targets of KaeA, 5′ UTRs of three genes coding for core histone proteins were also identified.

It seems likely that KaeA, as a chromatin-associated protein, might directly affect the transcription of genes identified in ChIP-Seq analysis. Some of them were also selected in transcriptomic analysis (Appendix A), confirming that KaeA directly affects their transcription. Among them is a gene encoding bZIP transcription factor HapX, involved in iron homeostasis regulation and interaction with the CCAAT-binding core complex [38,39]. However, not all genes identified in ChIP-Seq were also selected in transcriptomic analysis. This may be due to the fact that the *kaeA* deletion strain was not used for the transcriptome analysis, the *kaeA19* and *kaeA25* mutants were. These mutations lead to slight changes on the protein surface [22] that may only affect interactions with some protein partners, and hence the expression of only some genes directly regulated by KaeA.

Although KaeA is a chromatin-associated protein, it is not a DNA-binding protein, and its interactions with chromatin most probably occur through interactions with other proteins that directly bind to DNA or are part of DNA-binding complexes. Therefore, an important point in defining the role of KaeA in the regulation of transcription was identifying its protein partners other than KEOPS subunits. Proteomic analysis performed with the strain expressing KaeA-TEV-GFP fusion identified three proteins, interaction with which could explain the effect of KaeA on the expression level of many genes. These are AN2181, a homologue of yeast Toa2p, a small subunit of general transcription factor TFIIA involved in the initiation of transcription by RNA polymerase II, and two subunits of general transcriptional co-repressor, RcoA and SsnF, homologues of yeasts Tup1p and Ssn6p. It is worth pointing out that the interaction with RcoA/SsnF co-repressor was detected only in samples not treated with DNase, indicating that chromatin is needed for their occurrence (Table 2). In *A. nidulans*, RcoA/SsnF co-repressor is involved in the regulation of fungal growth and development and, to lesser extent, carbon regulation [40,41], and was shown to be essential for the maintenance of the closed chromatin structure of some promoters [42]. Yeast Tup1p/Ssn6p co-repressor is recruited to target genes by interactions with DNA-binding repressor proteins. Repression occurs via several mechanisms, such as interactions with histones [43], the recruitment of histone deacetylases (HDACs) [44], and interactions with the RNA polymerase II mediator subcomplex [45].

To function as a chromatin-associated regulatory protein, KaeA must be transported into the nucleus. Bioinformatics analysis predicted that KaeA is a cytoplasmic protein without any obvious nuclear localization signal (data not shown). However, the absence of an NLS sequence does not preclude transport into the nucleus. Proteins lacking this sequence can be transported into the nucleus in a complex with a protein that has the sequence. Such a mechanism was described, for example, for *Neurospora crassa* VeA and VelB proteins, subunits of the *velvet* regulatory complex. VeA is imported into the nucleus by KapA importin; in addition, it supports the nuclear transport of VelB, which lacks NLS [46]. It is possible that KaeA is transported into the nucleus in a complex with PipA, the other KEOPS subunit, homologue of yeast Bud32p. PipA includes NLS and was shown to be present in the nucleus [47]. Finally, KaeA transport to the nucleus and its interaction with other proteins may depend on post-translational modifications of the protein. It is very likely that one such modification is phosphorylation, which is known to regulate the cytoplasm/nuclear localization of transcription factors, as in the case of Mig1p carbon regulator of *S. cerevisiae* [48]. Proteomic analysis showed that KaeA interacts with protein kinases (CmkA, PkaA/PkaR, PkcB) and putative protein phosphatase (Table 2). Moreover, bioinformatics analysis predicted 31 potential phosphorylation sites, including some specific for protein kinases A and C (Appendix A). Altogether, these data suggest that KaeA may undergo phosphorylation and dephosphorylation reactions that affect its interaction with other proteins and/or nuclear localization.

In conclusion, our results indicate that KaeA is a protein with a dual function (Figure 3). We postulate that, in addition to its primary role, which is participating in t6A_37_ tRNA modification, KaeA has additional functions in the nucleus, taking part in the regulation of gene expression. As KaeA is a chromatin-associated protein but not a DNA-binding protein, its interactions with chromatin are most probably indirect and mediated via protein–protein interactions, i.e., via proteins that directly bind to DNA or are part of DNA-binding complexes. The influence of KaeA on the expression of many genes most likely results from its interactions with general transcription factor TFIIA and general co-repressor RcoA/SsnF. Possible interactions with other factors, including specific ones, cannot be ruled out. Our findings open up worthwhile research opportunities, as KaeA seems to be an attractive candidate as a model protein for multiple-function, linking translation and transcription regulation.

## 4. Materials and Methods

### 4.1. A. nidulans and S. cerevisiae Strains, Growth Conditions, and Transformation

*pyrG89*; *argB2*; *pabaB22*, *nkuA*Δ*::argB*; *riboB2* (1) was used as a recipient strain in transformation experiments and as a control strain in Western and proteomic analysis. *proA6*, *kaeA19*, *pabaA12*, *biA1* (2) and *proA6*, *kaeA25*, *adF9*, *yA2*, *phenA2* (3) (previously called *suD19pro* and *suD25pro*, respectively [22]) were used for primer extension and transcriptomic analysis. *proA6, pabaA9, biA1* (4) and *proA6, adF9, yA2, phenA2* (5) were used as control *kaeA^+^* strains. *kaeA::*[*TEV-GFP-A.f. pyrG], pyrG89; argB2; pabaB22, nkuAΔ::argB; riboB2* (6) was used in proteomic analysis. *kaeA::*[*HA-A.f. pyrG], pyrG89; argB2; pabaB22, nkuAΔ::argB; riboB2* (7) was used for ChIP-Seq analysis. *kaeA::*[*GFP*-*A.f*. *pyrG*], *pyrG89*; *argB2*; *pabaB22*, *nkuA*Δ*::argB*; *riboB2* (8) was used as a recipient strain in transformation experiments to obtain strains (6) and (7)*. yA1*, *pabaA1*, *areA::HA*, *argB::*[*fmdS-lacZ*] (9) was used as a control in ChIP-Seq experiment design (kind gift of Richard Todd) [49].

Transformation of *A. nidulans* was performed as described in [50]. For primer extension, transcriptomic, RT-qPCR, proteomic, and ChIP-Seq analyses, *A. nidulans* mycelia were grown on minimal medium (10 mM nitrate and 1% glucose) for 8–10 h at 37 °C, harvested, washed with cold distilled water, and frozen at −80 °C. 

*S. cerevisiae* haploid *kae1*Δ strain was obtained by sporulation and tetrad dissection of EJP51 diploid strain MATa/MAT*α: leu2*, *ura3*, *his3*, *ade2*, *trp1*, *KAE1/leu2*, *ura3*, *his3*, *ade2*, *trp1*, *kae1*Δ*::kan**MX6* (kind gift of Rolf Sternglanz) [7] and used in primer extension as an experimental control. Yeast *kae1*Δ and wild-type control strains were grown at 30 °C in YPD medium to OD_600_ = 1.5, harvested, washed with cold distilled water, and frozen at −80 °C.

### 4.2. Transcriptomics

For transcriptomic analysis, total RNA was isolated from *kaeA19*, *kaeA25*, and the respective control strains, as described by Schmitt et al., [51]. For each strain, three independent biological experiments were performed. Total RNA samples were treated with RQ1 RNase-Free DNase I (Promega, Madison, WI, USA) and ribosomal RNA was depleted with Epicenter Ribo-Zero™ Magnetic Gold Kit (Yeast) (Illumina, San Diego, CA, USA). The sample quality and success of depletion were assessed using an Agilent 2100 Bioanalyzer (Agilent Technologies, Santa Clara, CA, USA) and Agilent RNA 6000 Nano and Pico kits, respectively.

RNA-Seq libraries were prepared using the Epicentre ScriptSeq™ v2 RNA-Seq Library Preparation Kit (Illumina, San Diego, CA, USA). In this process, 17.5 ng of rRNA-depleted RNA, according to quantification by Bioanalyzer, was used as input, and following 10 cycles of amplification, libraries were purified using AMPure XP beads (Beckman Coulter, Brea, CA, USA). Libraries were quantified using a Qubit 2.0 fluorimeter (Invitrogen ™, Thermo Fisher Scientific, Waltham, MA, USA) and their size distribution and quality was assessed using the Bioanalyzer and Agilent High Sensitivity DNA kit (Agilent Technologies, Santa Clara, CA, USA). The quantity of each library was assessed by qPCR using KAPA Library Quantification Kit (Kapa Biosystems, Wilmington, MA, USA) on a Roche Light Cycler LC480II (Roche Diagnostics, Mannheim, Germany). The libraries were sequenced on the Illumina NextSeq 500 instrument using NextSeq High Output kit v1 (2 × 75 bp paired-end) (Illumina, San Diego, CA, USA).

Raw single reads were subjected to sequence quality control using FastQC v. 0.11.5 (http://www.bioinformatics.babraham.ac.uk/projects/fastqc/ accessed on 1 July 2016). The analysis was performed according to the RNA-Seq tutorial (https://www.bioconductor.org/packages//2.10/data/experiment/vignettes/RnaSeqTutorial/inst/doc/RnaSeqTutorial.pdf accessed on 1 July 2016). Paired-end reads were aligned to the *A. nidulans* reference genome sequence A_nidulans_FGSC_A4 version “s09-m04-r07”, downloaded from the *Aspergillus* Genome Database (AspGD). Alignment was conducted using TopHat v.2 [52] and Bowtie2 [53]. Mapped reads were filtered to retain only those where both reads of the pair (R1 and R2) aligned in the correct relative orientation. These were used to estimate expression levels for annotated loci by counting fragments. Fragment counting was conducted using ShortRead [54]. Alignment data are summarized in Appendix A. The mapped reads were analyzed using Python, with Pandas and Seaborn modules [55]. Fragment counts were used to assess differential gene expression, after normalization for library size, using the R package edgeR [56,57]. Gene expression was regarded as significantly changed if fold change was >2 and adjusted *p*-value was <0.05. Protein function annotation was performed using the AspGD [58], Fungal & Oomycete Informatic Resources (FungiDB) [59,60], and Saccharomyces Genome Database (SGD) [61]. Gene Ontology (GO) enrichment analysis was performed using the software available on FungiDB (Biological Process, *p*-value < 0.05). The RNA-Seq data discussed in this publication were deposited in NCBI’s Gene Expression Omnibus [62] and are accessible under GEO Series accession number GSE206830.

### 4.3. RT-qPCR Analysis

For RT-qPCR analysis, total RNA from *kaeA25*, *kaeA19*, and the respective control isogenic strains was isolated, as described by Schmitt et al., [51]. RNA was treated with DNase I (Roche Diagnostics, Mannheim, Germany), phenol-chloroform extracted and ethanol precipitated. The sample quality and concentration were assessed using an Agilent 2100 Bioanalyzer and Agilent RNA 6000 Nano kit (Agilent Technologies, Santa Clara, CA, USA). Only RNA samples with RNA integrity number (RIN) value not lower than 9.0 were used for reverse transcription. cDNA was synthesized from 2 μg of total RNA using SuperScript^®^ III Reverse Transcriptase (Invitrogen™ Thermo Fisher Scientific, Waltham, MA, USA), and a mixture of oligo-dT and random hexamer primers, in accordance with the manufacturer’s protocol. RT-qPCR was conducted using the LightCycler^®^ 480 II System (Roche Diagnostics, Mannheim, Germany) with pairs of primers specific for selected genes (Appendix A) and LightCycler^®^480 SYBR Green I Master Mix (Roche Diagnostics, Mannheim, Germany). For each strain, 3 biological repeats were analyzed, with 2 technical repeats in each case. The efficiency (E) and specificity of each pair of primers were tested in RT-qPCR reactions using 6-point standard curves of 5-fold diluted cDNA of the control strain. The E value for all primer pairs used was in the range of 1.93–2.00. Cp values were calculated using LightCycler^®^480 Software v.1.5 (Roche Diagnostics, Mannheim, Germany), based on the second derivative maximum method. Cp values were normalized using amplicon for 18S rRNA as an endogenous control. Differential expression was assessed using the *t*-test with Benjamini–Hochberg correction, using the wild type as a reference.

### 4.4. Plasmids for Aspergillus Transformation

#### 4.4.1. Golden Gate Cloning

The Golden Gate cloning system was used to obtain a plasmid expressing KaeA-GFP fusion. A plasmid comprising a *kaeA::*[*GFP-A.f.pyrG*]-*kaeA-DR* cassette (where *A.f.pyrG*, the *Aspergillus fumigatus pyrG* gene, was used as a selection marker, and *kaeA-DR* refers to *kaeA* downstream region) was constructed using the Golden Gate cloning procedure [63]. All the plasmids and DNA fragments were purified with a High Pure PCR Product Purification Kit (Roche Diagnostics, cat. no. 11732668001, Mannheim, Germany). Fragments for cloning were amplified using specific primers (Appendix A). The Golden Gate cloning reaction was performed according to the standard protocol, with 75 ng storage vector, 75 ng target vector (pET28), 40 ng of each insert, 1 µL T4 DNA ligase buffer (Roche Diagnostics, Mannheim, Germany), 0.5 µL BveI (10 U/ µL; New England Biolabs, Ipswich, MA, USA;), and 0.75 µL T4 DNA ligase (0.75 U/ µL; Roche Diagnostics, Mannheim, Germany) added to the tube. The PCR was carried out according to the following program: 37 °C for 2 min and 16 °C for 5 min over 50 cycles, 37 °C for 5 min, 5 °C for 5 min, and 80 °C for 5 min.

Then 20 µL of the Golden Gate reaction mixture was used to transform GC10™ Competent Cells (Sigma-Aldrich, Saint Louis, MO, USA) and transformants were selected on an LB + kanamycin (100 µg/mL) plate. The linear fragment comprising *kaeA::*[*GFP-A.f.pyrG*]-*kaeA-DR* cassette was cut out of the plasmid and used for transformation of the *A. nidulans*
*nkuA*Δ, *kaeA^+^* recipient strain (1), giving the strain expressing KaeA-GFP fusion (8).

#### 4.4.2. Overlap Extension PCR Cloning

Overlap extension PCR cloning was used to obtain strains expressing KaeA-TEV-GFP (6) and KaeA-HA (7) fusions [64]. Fragments were amplified using specific primers (Appendix A). The first-step primers contained 30 nucleotides complementary to the insert sequence (coding for TEV protease recognition site or HA-tag) and 50 nucleotides complementary to the target vector sequence (vector comprising *kaeA::*[*GFP-A.f.pyrG*]-*kaeA-DR* cassette; Section 4.4.1). The first-step reaction was performed using the Expand Long Template PCR System polymerase (Roche Diagnostics, Mannheim, Germany) in accordance with the manufacturer’s recommendations. In the second step, the PCR product obtained from the first step was used as a primer. A 250-fold molar excess of PCR product from the first step, 0.6 uL 10 mM dNTP, 1.25 uL 10× reaction buffer, and 0.4 uL of Expand Long Template PCR System polymerase were added to 10 ng of the target vector, for a final volume of 12.5 uL. The PCR reaction was carried out according to the following program: 98 °C for 5 min; 30 cycles of 98 °C for 30 s, 60 °C for 30 s, and 72 °C for 1.5 min per kb; and 72 °C for 10 min. After the completion of the reaction, 10 U DpnI (Thermo Fisher Scientific, Waltham, MA, USA) was added directly to the reaction mixture, and the sample was incubated for 1 h at 37 °C. GC10™ Competent Cells (Sigma-Aldrich, Saint Louis, MO, USA) were transformed with the reaction mixture according to the manufacturer’s protocol. The resulting plasmids were used for transformation of the *A. nidulans*
*nkuA*Δ*, kaeA^+^* recipient strain (1), giving strains (6) and (7), expressing KaeA-TEV-GFP and KaeA-HA fusion proteins.

### 4.5. Purification of Proteins Interacting with KaeA

For proteomic analysis, *A. nidulans* strain (6) expressing KaeA-TEV-GFP fusion protein and control isogenic strain (1) were grown. Four independent biological experiments were performed for each strain, two of them with the addition of DNase during the purification procedure (see below).

Then 20 g of frozen mycelium was ground in a cooled mill with the addition of dry ice, suspended in 25 mL of buffer (40 mM HEPES, pH 8.0; 150 mM NaCl; 1 mM DTT; protease inhibitors: 2 mM benzamidine, 1 μM leupeptin, 2 μM pepstatin, 4 μM chymostatin; 2 mM PMSF) and ultracentrifuged at 20,000 rpm for 15 min at 4 °C. The supernatant was transferred to a new tube and ultracentrifuged at 32,000 rpm for 1 h 15 min at 4 °C.

Supernatant was dialyzed in VISKING^®^ dialysis tubing 36/32 (Serva Electrophoresis, Heidelberg, Germany) against dialysis buffer IPP 150 (10 mM HEPES, pH 8.0; 150 mM NaCl, 1 mM PMSF; 1 mM DTT with 20% glycerol) for 2 h at 4 °C. Extract was transferred to a test tube with 100 µL of GFP-TrapA resin (ChromoTek, Planegg, Germany) previously washed with IPP 150. Subsequently, Triton™ X-100 to 0.1%, PMSF to 0.1 mM, protease inhibitors (benzamidine to 2 mM, leupeptin to 1 µM, pepstatin to 2 µM, chymostatin to 4 µM), 1/1000 volume of A + T1 RNase Cocktail™ (Ambion™, Thermo Fisher Scientific, Waltham, MA, USA), and, optionally, 2 U of DNase TURBO (Invitrogen™, Thermo Fisher Scientific, Waltham, MA, USA) were added. After 12 h of incubation with constant agitation at 4 °C, the extract was transferred to Poly-Prep^®^ Chromatography Columns (Bio-Rad Laboratories, Hercules, CA, USA). The resin was rinsed 3 times with 10 mL of IPP 150 and once with 10 mL of TEV protease buffer (20 mM Tris-HCl, pH 8; 500 mM NaCl; 10 mM imidazole; lysozyme 50 μg/mL). Proteins were eluted by incubation with 10 μg of TEV protease in 250 μL TEV protease buffer for 2 h at room temperature. Water was added to the eluate to a volume of 400 µL, and then 100 µL Pyrogallol Red (Sigma-Aldrich, Saint Louis, MO, USA) was added. The sample was vortexed, incubated for 20 min at room temperature, and centrifuged for 15 min at 12,000 rpm.

Precipitated proteins were subjected to MS analysis at the Laboratory of Mass Spectrometry, Institute of Biochemistry and Biophysics, Polish Academy of Sciences. Precipitated proteins were digested with trypsin. MS analysis was performed by LC-MS using a nanoAcquity UPLC system (Waters, Milford, MA, USA) coupled to an Orbitrap Elite (Thermo Fisher Scientific, Waltham, MA, USA). Peptide mixtures were applied in equal volumes of 20 μL to RP-18 pre-column (Waters) using water containing 0.1% FA as a mobile phase and then transferred to a nano-HPLC RP-18 column (internal diameter 75 µm; Waters) using ACN gradient (0–35% ACN for 160 min) in the presence of 0.1% FA at a flow rate of 250 nL/min. The mass spectrometer was operated in the data-dependent MS2 mode, and data were acquired in the m/z range of 300–2000.

Data were preprocessed with Mascot Distiller (version 2.5; Matrix Science, Boston, MA, USA) and searched with the Mascot software version 2.5 (Matrix Science) [65] against the *A. nidulans* database derived from FungiDB v. 44 [59,60]. The rest of the parameters were as follows: enzyme: trypsin; fixed modification: methylthio (C); variable modification: oxidation (M). Data were recalibrated offline, with typical peptide mass tolerance of 5 ppm and fragment mass tolerance of 0.01 Da. FDR was calculated using the Decoy database and false positive rate was kept below 1%. The mass spectrometry proteomics data were submitted to the ProteomeXchange Consortium [66] via the PRIDE [67] partner repository with dataset identifiers PXD034554 and 10.6019/PXD034554.

Proteomes of the strain expressing KaeA-TEV-GFP fusion protein and the control isogenic strain were compared. Proteins were selected that were not present in the control strain and were present at least twice in the strain expressing the KaeA fusion, including at least once with Mascot score > 70.

### 4.6. Primer Extension

RNA from yeast *kae1*Δ and wild-type control strains, and from *A. nidulans* strains (2), (3), (4), and (5) was isolated, as described by Schmitt et al., [51]. A fraction of small RNAs was purified using the mirVana™ miRNA isolation kit (Ambion™, ThermoFisher, cat. No. AM1560, Waltham, MA, USA). The RNA concentration was checked using a NanoDrop ND-2000 spectrophotometer (ThermoFisher).

A modified primer extension experiment was performed according to the procedure described for *S. cerevisiae* [7]. HPLC purified primers IletRNArev, ValtRNArev, and IletRNArev-yeast (Appendix A) were labeled at the 5′ end with T4 polynucleotide kinase (ThermoFisher, cat. No. EK0031) and [γ-32P] ATP (6000 Ci/mmol; Perkin Elmer, Waltham, MA, USA). Then 17 µL of ammonium acetate, 0.5 µL of GlycoBlue™ co-precipitant (15 mg/mL) (Invitrogen™, ThermoFisher, cat. No. AM9515), and 100 µL of 95% ethanol were added to the samples. Labeled primers were precipitated for 1 h at −20 °C and then centrifuged at 12,000 rpm for 15 min at 4 °C. Pellets were resuspended in 10 µL of water. Next, 1 pmol of primer and 2 μg of small RNA fraction in 5 µL First-Strand Buffer (Invitrogen™, ThermoFisher) were incubated for 3 min at 95 °C, followed by slow cooling to 37 °C. Subsequently, 5 µL of reaction mixture (0.5 mM dNTPs, 10 mM DTT, and 200 units of SuperScript™ III reverse transcriptase (Invitrogen™, ThermoFisher) in First-Strand Buffer) was added to the RNA–primer mixture. Elongation was performed for 50 min at 42 °C, followed by inactivation (15 min, 70 °C). Then, 10 µL of Gel Loading Buffer II (Ambion™, ThermoFisher) was added to each sample, which was run on a 15% denaturing polyacrylamide sequencing gel (0.4 mm). The gel was dried, and the samples were visualized with an FLA-9000 laser imager (GE Healthcare, Chicago, IL, USA).

### 4.7. Western Analysis

The frozen mycelia were resuspended in 1 mL of homogenization buffer (40 mM HEPES, pH 8.0; 150 mM NaCl; 1 mM DTT; 2 mM PMSF) and ground using an MP Biomedicals FastPrep^®^-24 bead beater (MP Biomedicals, Irvine CA, USA), (2 cycles, 60 s, 6 m/s). The extract was centrifuged 2 times at 12,000 rpm for 15 min at 4 °C.

SDS-PAGE was performed according to Sambrook and Russell [68], and the Western analysis was performed as described by Todd et al., for AreA-HA fusion [49]. In this step, 10 μL of the protein extract was mixed with Laemmli’s reagent (1:1) containing 5% β-mercaptoethanol, incubated for 10 min at 95 °C, and loaded onto a polyacrylamide gel (4% stacking gel and 12% separating gel). The PageRuler™ Prestained Protein standard (Thermo Fisher) was used. Electrophoresis was carried out in a Mini-Protean apparatus (Bio-Rad Laboratories) using TG buffer (25 mM Tris, 250 mM glycine, 0.1% SDS), for 2 h at 65 V. After the completion of electrophoresis, the gel was placed in the transfer buffer (50 mM Tris, 40 mM glycine, 0.0375% SDS, 20% methanol) for 5 min, and subsequently, proteins were transferred to Protran membrane (Whatman™, Sigma-Aldrich, Saint Louis, MO, USA) using a Trans-Blot apparatus (Bio-Rad Laboratories) for 1 h at 10 V.

The membrane was preincubated in 5% skim milk and 0.1% Tween in PBS buffer (Sigma-Aldrich, Saint Louis, MO, USA) for 1 h. Primary Anti-HA antibody (Roche Diagnostics, cat. no. 11583816001, Mannheim, Germany) at a 1:5000 dilution was added, and incubation was carried out for 12 h at 4 °C with constant agitation. The membrane was washed 3 times for 10 min in 0.1% Tween in PBS, and then incubated with secondary anti-mouse antibody (Sigma-Aldrich, cat. no. A9044, Saint Louis, MO, USA) at 1:10,000 dilution in the same buffer for 1 h at room temperature. Finally, the membrane was washed as above and the SuperSignal™ West Femto Maximum Sensitivity Substrate Chemiluminescence Kit (ThermoFisher, cat. no. 34094) was used to visualize the labeled antibodies, with detection by a CCD camera (Bio-Rad Laboratories).

### 4.8. ChIP-Seq Analysis

*A. nidulans* strain (7) expressing KaeA-HA fusion protein was used for ChIP-Seq analysis. Strain (9) expressing AreA-HA fusion protein was used as an additional control for chromatin immunoprecipitation (ChIP). Three independent biological experiments were performed.

ChIP was conducted according to Boedi and Strauss [69]. Incubation for 15 min with 1% formaldehyde was used for crosslinking, and then mycelia were ground in a mortar. Extracts were sonicated for 25 min using a Bioruptor 200 (Diagenode, Ougrée, Belgium) (2 min on/1 min off/power setting high). Anti-HA antibody (Roche Diagnostics, cat. no. 11583816001, Mannheim, Germany) was used for chromatin immunoprecipitation. Control qPCR reaction was conducted for samples from strain expressing AreA-HA fusion, using primers specific for bi-directional *niiA–niaD* promoter (Appendix A), as described by Berger et al., [25]. Input DNA before immunoprecipitation was used as a control to assess enrichment of specific DNA regions after immunoprecipitation.

ChIP-Seq was performed at the Next-Generation Sequencing Core Facility, Centre of New Technologies, University of Warsaw. ChIP-Seq libraries were prepared using KAPA HyperPrep Kit (Kapa Biosystems; #KR0961) and KAPA dual-indexed adapters (Kapa Biosystems; #KK8722) with 10 cycles of amplification. The size distribution was assessed using an Agilent 2100 Bioanalyzer and High Sensitivity DNA Kit (Agilent Technologies; #5067–1513, Santa Clara, CA, USA). The concentration of the libraries was determined by qPCR using a Kapa Library Quantification Kit (Kapa Biosystems; #KK4824) and Roche LC480II cycler (Roche Diagnostics, Mannheim, Germany) following the manufacturer’s protocols. Sequencing was performed on an Illumina NovaSeq 6000 instrument using a NovaSeq 6000 S2 Reagent Kit (100 cycles) (Illumina, cat. no. 20012862, San Diego, CA, USA) in 2 × 100 cycles pair-end reading mode, following the manufacturer’s recommended procedure.

Raw reads were trimmed with the use of Trimmomatic software [70]. All reads with a quality less than 20 (phred+33) in the four-nucleotide window were trimmed. Reads shorter than 50 nucleotides were discarded. Trimmed reads were mapped to the reference genome with the use of BWA [71]. All the samples were mapped separately to the Aspergillus_nidulans.ASM1142v1 reference genome derived from Ensembl Fungi [72]. Optically duplicated reads were identified and removed with the use of the Picard tool. Alignment data are summarized in Appendix A. Peak calling was performed in R software [73] with the use of system PipeR, CHIPpeakAnno, GenomicFeatures, ChIPseeker, and Biostrings packages. All biological replicates (fastq files) were merged into one file for input and immunoprecipitated (IP) samples. Peak calling with input/IP samples was performed with the use of the MACS2 callpeak algorithm [74]. Differential binding analysis between control and treatment was performed with the use of the edgeR package [56,57]. Among all detected peaks, 104 differed with statistical significance *p* < 0.05 (FDR corrected). Finally, 78 genes were identified. Annotation with GO, Pfam, and KEGG databases was performed with the use of Interproscan [75].

The ChIP-Seq data discussed in this publication were deposited in the NCBI Gene Expression Omnibus database and are accessible under GEO series accession number GSE206874.

### 4.9. Bioinformatics Analysis of KaeA Protein

DeepLoc 2.0 [76] was used to predict the subcellular localization of KaeA, and NetPhos-3.1 [77,78] was used to predict KaeA phosphorylation sites.

## Figures and Tables

**Figure 1 ijms-23-11138-f001:**
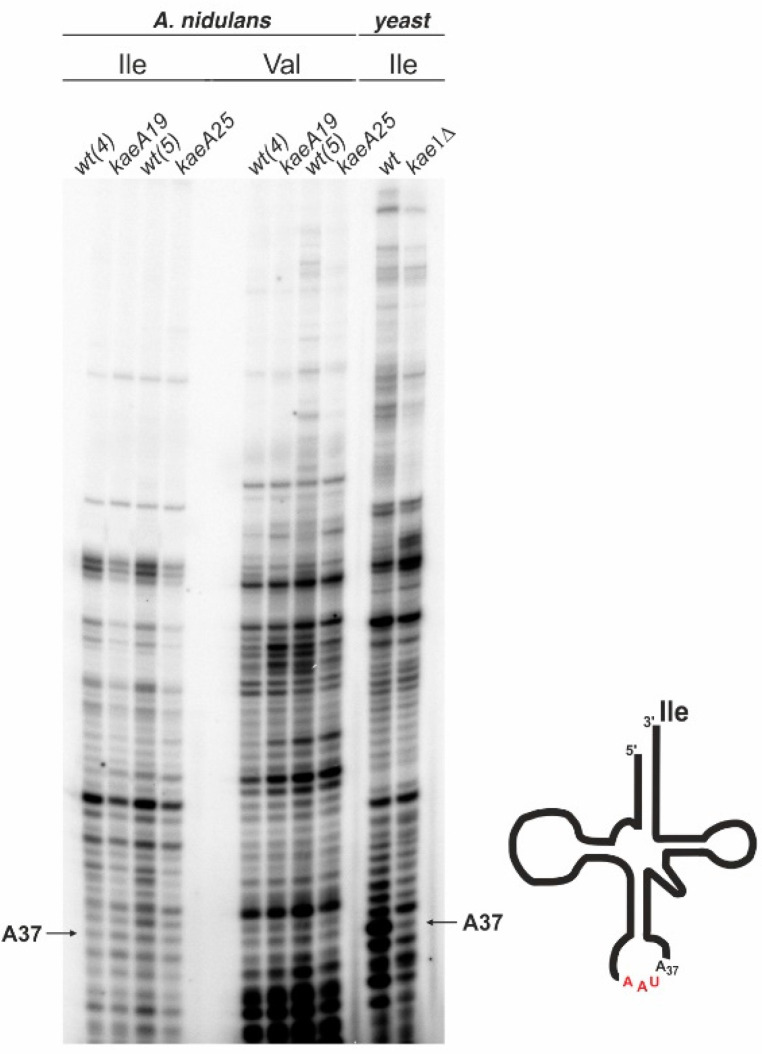
Primer extension analysis of tRNA^Ile^ and tRNA^Val^ in *A. nidulans* and *S. cerevisiae.* A fraction of small RNAs was isolated from *kaeA19*, *kaeA25*, *kae1*Δ mutant, and respective control strains. The position of the modified t6A_37_ nucleotide is indicated.

**Figure 2 ijms-23-11138-f002:**
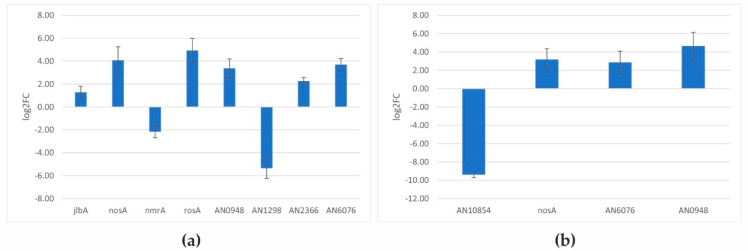
Quantitative transcriptional analysis of selected genes differentially expressed in (**a**) *kae19* and (**b**) *kaeA25* mutant strains. Relative expression in *kaeA* mutants in comparison with control isogenic wild-type strain was calculated by RT-qPCR analysis. FC, fold change in *keaA19(25)*/wt.

**Figure 3 ijms-23-11138-f003:**
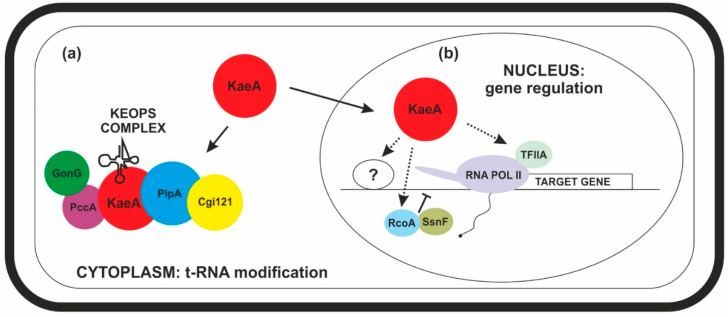
KaeA, a protein with a dual function. (**a**) Primary role of KaeA: participation in t6A_37_ tRNA modification. (**b**) Additional nuclear function of KaeA: regulation of gene expression.

**Table 1 ijms-23-11138-t001:** Differentially expressed genes coding for regulatory proteins selected for RT-qPCR analysis.

Gene AN Number	Gene Name	Yeast Homologue Gene Name ^1^	Protein Function ^2^
*AN5170*	*rosA*	*UME6*	Putative Zn(II)2Cys6 transcription factor; negative regulator of sexual development. Homologue of yeast Ume6p, encoding histone Rpd3p deacetylase subunit, general regulator of chromatin structure and transcriptional repressor.
*AN1848*	*nosA*	*UME6*	Putative Zn(II)2Cys6 transcription factor; negative regulator of sexual development. Homologue of yeast Ume6p, encoding histone Rpd3p deacetylase subunit, general regulator of chromatin structure and transcriptional repressor.
*AN1812*	*jlbA*	–	bZIP transcription factor involved in amino acid biosynthesis.
*AN8168*	*nmrA*	–	Co-repressor of general nitrogen regulator AreA
*AN0948*	–	*GCN20*	Protein with predicted ATPase activity. Homologue of yeast GCN20p, positive regulator of Gcn2p kinase; forms a complex with Gcn1p; proposed to stimulate Gcn2p activation by an uncharged tRNA.
*AN1298*	–	*RTG3*	Homologue of yeast Rtg3p, bHLH/Zip transcription factor for retrograde (RTG) and TOR pathways.
*AN6076*	–	*SWR1*	Protein with helicase domain of SNF2 type. Homologue of yeast Swr1p, Swi2/Snf2-related ATPase; catalytic subunit of SWR1 complex, which exchanges histone variant H2AZ (Htz1p) for chromatin-bound histone H2A.
*AN10854*	–	*SNF4*	Putative serine-threonine protein kinase. Homologue of yeast Snf4p, activating subunit of AMP-activated Snf1p kinase complex; activates glucose-repressed genes, represses glucose-induced genes.
*AN2366*	–	*–*	Putative trypsin-like protease involved in proteolytic cleavage of NmrA, a regulatory protein involved in repression of nitrogen metabolism.

^1^ Dash (–) indicates lack of yeast homologue. ^2^ According to FungiDB and SGD.

**Table 2 ijms-23-11138-t002:** Potential protein partners of KaeA.

Gene AN Number	Gene Name	Protein Function (from FungiDB)	Yeast Homologue Gene Name	Protein Function of Yeast Homologue (from SGD)	DNAse * +/−
AN2845	*pccA*	Putative component of EKC/KEOPS complex	*PCC1*	Component of KEOPS complex	+
AN2513	*pipA*	Putative component of EKC/KEOPS complex	*BUD32*	Protein kinase, component of KEOPS complex	+
AN11910	*cgi121*	Putative component of EKC/KEOPS complex	*CGI121*	Component of KEOPS complex	−
AN11901	*gonG*	Putative component of EKC/KEOPS complex	*GON7*	Component of KEOPS complex	+
AN6505	*rcoA*	WD40 repeat protein	*TUP1*	Chromatin-silencing transcriptional regulator; forms complex with Cyc8p, involved in establishment of repressive chromatin structure	−
AN12054	*ssnF*	Orthologue of *S. cerevisiae* Ssn6p	*CYC8*	Chromatin-silencing transcriptional regulator; acts together with Tup1p	**−**
AN2181	** *–* **	Orthologue(s) have RNA polymerase II transcription factor activity, TBP-class protein binding	*TOA2*	TFIIA small subunit; involved in transcriptional activation, acts as anti-repressor or co-activator	+
AN2412	*cmkA*	Calcium/calmodulin-dependent protein kinase A	*CMK2*	Calmodulin-dependent protein kinase	−
AN6305	*pkaA*	cAMP-dependent protein kinase catalytic subunit	*TPK2*	Catalytic subunit of cAMP-dependent protein kinase	+
AN4987	*pkaR*	Putative protein kinase A (PKA) regulatory subunit	*BCY1*	Regulatory subunit of cyclic AMP-dependent protein kinase PKA	+
AN5973	*pkcB*	Protein with similarity to protein kinase C	*YPK1*	Serine/threonine protein kinase	+
AN1545	*–*	Putative regulatory subunit of protein phosphatase 2A (PP2A)	*CDC55*	Non-essential regulatory subunit B of protein phosphatase 2A (PP2A)	+

Note: for complete list of proteins interacting with KaeA, see Appendix A. * Occurrence in DNase-treated sample.

**Table 3 ijms-23-11138-t003:** Chromatin regions interacting with KaeA.

Gene AN Number	Gene Name	Protein Function (from FungiDB)	Yeast Homologue Gene Name *	Protein Function of Yeast Homologue (from SGD)	Chromatin Region Interacting with KaeA
AN0733	*hhtA*	Histone H3	*HHT1*	Histone H3; core histone protein required for chromatin assembly, part of heterochromatin-mediated telomeric and HM silencing	5′ UTR
AN0734	*H4.1*	Histone H4.1	*HHF2*	Histone H4; core histone protein required for chromatin assembly and chromosome function; contributes to telomeric silencing	5′ UTR
AN3468	*H2A.X*	Histone H2A	*HTA1*	Histone H2A; core histone protein required for chromatin assembly and chromosome function	5′ UTR
AN1007	*niiA*	Putative nitrite reductase with predicted role in nitrogen metabolism	*–*	–	Promoter region
AN2436	*aclB*	Putative ATP citrate synthase with predicted role in TCA cycle	*–*	–	5′ UTR
AN2435	*aclA*	Putative ATP citrate synthase with predicted role in TCA cycle	*–*	–	5′ UTR
AN3675	*cpcA*	Transcription factor of c-Jun family of Gcn4p-like transcriptional activators	*GCN4*	bZIP transcriptional activator of amino acid biosynthetic genes; responds to amino acid starvation	5′ UTR
AN4159	*glnA*	Putative ammonium glutamate ligase with predicted role in glutamate and glutamine metabolism	*GLN1*	Glutamine synthetase (GS); synthesizes glutamine from glutamate and ammonia; with Glt1p, forms secondary pathway for glutamate biosynthesis from ammonia	5′ UTR
AN4376	*gdhA*	Putative glutamate dehydrogenase associated with NADP	*GDH1*	NADP(+)-dependent glutamate dehydrogenase	5′ UTR
AN5823	*sidA*	L-ornithine N5-monooxygenase	*–*	–	5′ UTR
AN7463	*meaA*	Main ammonium transporter in *A. nidulans*	*MEP3*	Ammonium permease with high capacity and low affinity	5′ UTR
AN8041	*gpdA*	Glyceraldehyde-3-phosphate dehydrogenase with predicted role in gluconeogenesis and glycolysis	*TDH3*	Glyceraldehyde-3-phosphate dehydrogenase (GAPDH), isozyme 3; involved in glycolysis and gluconeogenesis	5′UTR
AN8251	*hapX*	bZIP transcription factor	–	–	exon

Note: for the complete list of chromatin regions interacting with KaeA, see Appendix A. * Dash (–) indicates lack of yeast homologue.

## Data Availability

The RNA-Seq transcriptomic data discussed in this publication have been deposited in the NCBI Gene Expression Omnibus database [62] and are accessible under GEO series accession number GSE206830 (https://www.ncbi.nlm.nih.gov/geo/query/acc.cgi?acc=GSE206830). The ChIP-Seq data discussed in this publication have been deposited in NCBI’s Gene Expression Omnibus [62] and are accessible under GEO series accession number GSE206874 (https://www.ncbi.nlm.nih.gov/geo/query/acc.cgi?acc=GSE206874). The mass spectrometry proteomics data discussed in this study are openly available from the ProteomeXchange Consortium [66] via the PRIDE [67] partner repository at Project DOI: 10.6019/PXD034554, with dataset identifier PXD034554.

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
