# Peer review of "Nuclear Functions of KaeA, a Subunit of the KEOPS Complex in Aspergillus nidulans"

_ijms, 2022, doi:10.3390/ijms231911138_

Round 1
Reviewer 1 Report
The work Nuclear functions of KaeA, the subunit of KEOPS complex in Aspergillus nidulans by Gawlik et al., is aimed at characterizing the role of the KaeA component of the tRNA modified complex in A. nidulans. It was previously described that KaeA plays an important role in modification of tRNA(t6A37) that results in regulation of translation initiation of different genes. In this article the authors focus the study on its putative effect at the level of transcription and translation. Using advanced molecular gene tools the authors found that KaeA affect the expression of numerous genes. Also, KaeA is a component of the chromatine complex and therefore may regulated gene expression. This is a key aspect of the article and the specificity of binding of KaeA to other proteins should be clarified. In general, the article is scientifically interesting and is well prepared but some obscure points are indicated below. The manuscript reviewer lacks line numbers that should be included to facilitate indication of possible changes in the text.
Specific Comments
1. The manuscript has not line numbers, therefore it is difficult to evaluate specific sentences. In the last part of the Introduction, the authors state “We postulated that the expression of some genes, e.g. arginine catabolism genes, may be directly regulated at the level of transcription by the KaeA, although, most probably several other observed pleiotropic effects of the kaeA19/kaeA25 mutations were indirect”. Since the authors indicate in the Introduction that there are maybe many indirect effects on gene expression in mutants in KaeA, they should explain later in the Results and the Discussion which is the final conclusion that they obtain after the analysis of hundreds of genes that are affected by the KaeA mutation.
2. In section 2.4 the authors describe the interaction of KaeA with chromatine using the ChIp-Seq tool. The authors indicate “The difficulty in conducting this experiment was that KaeA is not a typical DNA binding protein and no sequence to which it binds was known. Usually, the ChIP-Seq experiment is carried out for proteins to which at least one DNA target sequence is known. Such a sequence is used in a qPCR control reaction to confirm that ChIP experiment was done correctly.”. This experiment is critical to advance in the understanding the role of KaeA in relation to chromatine rearrangements and gene expression, the author should emphasize that the results obtained by comparison of the mutants with the wild type strain avoid the experimental limitation that they indicate.
3. At the end of the Results section and in Figure 3 the authors analyse possible KaeA binding sites in the genome and conclude that there are KaeA four possible binding sites that occasionally are coincident with the binding sites of other proteins. Since the authors describe up to 78 genes that tentatively are bound by KaeA the analysis of conserved binding sites is important in this article. However, the authors dedicate only a few lines to this point. In conclusion, it seems that a significant part of the 78 genes contain binding sequences for other regulatory proteins. This should be clarified since may be this are not specific binding sequences that may explain if KaeA interact with other regulatory proteins and therefore these other regulatory proteins identify those sequences rather than KaeA.
4. In the Discussion the authors give an interesting vision of the regulatory effect of KaeA in respect to expression of numerous genes. This should be reinforced to clarify the specificity of the regulation effect versus the indirect effect
5. The authors indicate in several places in the text that KaeA works as a nuclear protein binding to DNA. However, for this purpose KaeA need to enter into the nucleus and nuclear transport proteins are required; KaeA probably suffer specific protein modification. This subject is not addressed by the authors and should be included in the manuscript.
Author Response
We thank the Reviewer for detailed comments and suggestions.
Regarding line numbering in the manuscript, we would like to apologize for the inconvenience. The text was prepared using the IJMS template file with the line numbering. However, we have noticed that the line numbering is not visible on some of our computers. We were unable to explain why this problem arises, so for the convenience of the Reviewer, in addition to the doc format file, we attach a pdf file with visible line numbering.
Regarding protein partners of KaeA, proteins were selected that were not present in the control strain and were present at least twice in the strain expressing the KaeA fusion, including at least once with Mascot score >70, which is a standard value for this type of analysis. To show these data more precisely, 4 columns with Mascot score values ​​in each of the four independent biological experiments were added to Table S6. However, it should be emphasized that all these data have been deposited in the ProteomeXchange database, as indicated in the "Data Availability Statement" - l.731-734
As the Reviewer suggested, the direct and indirect effect of KaeA on the transcription of genes whose expression levels changed in kaeA mutants, was thoroughly discussed in the relevant sections of the Results (l. 152-164 and 170-177) and Discussion (l. 309-328).
The ChIP-Seq experiment was conducted and analyzed in a standard manner. As standard, it does not compare samples from the two strains, test and control, only chromatin from the test strain before immunoprecipitation (input DNA) and after immunoprecipitation. The strain expressing AreA::HA was only used to validate the chromatin immunoprecipitation step. This is further explained in the relevant sections of the Results (l. 224-243).
We postulate that KaeA is a chromatin - associated protein, not a DNA binding protein, and that interactions of KaeA with chromatin are indirect and occur through interactions with DNA-binding proteins or DNA-binding complexes. We agree with the Reviewer that this was unclear in several parts of the manuscript. Taking this into account, part 2.4 of the Results has been revised. In the revised manuscript, we are writing about chromatin regions interacting with KaeA, not genes interacting with KaeA. A new column has also been added to Table 3 ("Chromatin region interacting with KaeA"). In the first version of the manuscript this data were shown only in Table S8. The respective parts of Disscussion, describing the indirect interaction of KaeA with chromatin, have also been revised (l. 340 -385). We also decided to remove Figure 3, which may be confusing to readers, suggesting that we had identified KaeA's DNA binding site. We postulate that KaeA is a chromatin - associated protein, not a DNA binding protein. The results of motif search analysis are preliminary results for further research and will be published in a subsequent paper.
According to the Reviewer suggestions, a bioinformatic analysis of KaeA protein localization and phosphorylation was performed, and its results were shown and discussed in the the Mat/Met Section 4.9, Table S11 and Discussion (l.386-405).
Finally, the conclusions (l.406-417) were revised respectively, the text has been corrected to remove ambiguities and professional linguistic proofreading was performed.
Reviewer 2 Report
KaeA is a component of the cytoplasmic KEOPS complex, which comprises five different subunits and catalyses the formation of the modified nucleoside N6-threonylcarbamoyl adenosine at position 37 (t6A37) of tRNAs decoding ANN codons. There exists evidence that, in addition to this function, KaeA may have a role in transcriptional regulation. The authors investigated the Aspergillus nidulans kaeA19 mutant lacking the N-terminal four amino acids of KaeA and the kaeA25 mutant, in which the C-terminal four amino acids are replaced by a single amino acid. Although the t6A37 modification seems unaffected in both mutants, they grow considerably more slowly than wild type A. nidulans. Furthermore, compared to wild type, they reveal significantly altered transcriptomes as shown via the Illumina technique. For some elected differentially expressed genes, the authors confirmed the change in transcription via RT-qPCR.
To identify proteins potentially interacting with KaeA, the authors generated a KaeA-GFP fusion protein, in which the KaeA and GFP components can be separated by TEV cleavage. Using an elegant assay employing GFP trap chromatography and DNase or not, they identified 89 proteins interacting with KaeA. Reassuringly, among those proteins, there were all components of the KEOPS complex. In addition, they identified a number of protein kinases, a putative protein phosphatase, translation initiation factors, and several proteins involved in transcription regulation. Remarkably, all candidates known to have a function in transcription regulation and/or chromatin binding were only identified as putative KaeA interaction mates in the absence of DNase, i.e., in the presence of chromatin. Ultimately, via ChIP-Seq using a KaeA-HA construct, the authors identified 78 genes which interact with KaeA. Since KaeA is not a DNA-binding protein, these interactions most probably occur indirectly via further proteins, which directly bind to DNA.
The presented work appears technically sound, the results are interesting and significant and clearly provide a valuable basis to analyse the role of chromatin-associated KaeA in transcription regulation in more detail. However, the language of the manuscript is definitely in need of improvement. I strongly suggest some professional editing before publication.
Author Response
As suggested by the Reviewer 2, the text has been corrected to remove ambiguities, and professional linguistic proofreading was performed. Additionally, a bioinformatic analysis of KaeA protein localization and phosphorylation was performed, and its results were shown in Table S11 and discussed at the end of Discussion. We would like to thank the Reviewer for accurately describing KaeA as "chromatin - associated protein", which was used in the text.
Round 2
Reviewer 1 Report
In this modified version the authors have changed considerably the focus of the article and the interpretation of the results. they now emphasize the interpretation that KaeA interact with different proteins at the chromatin level and therefore affect the expression of many genes. This is more a consistent with the observation in which the authors find hundreds of genes affected by KaeA, however the authors do not reply to some of the reviewer questions.
1. As indicated by the reviewer in point 1 of the previous evaluation, the authors postulate, the KaeA bind to some genes, such as the arginine catabolic genes, but they do not clarify this point. Is this based in real experimental results? or it is simply a postulate of the authors that is not supported by the present evidence?. Please clarify
2. In the modified version, lines 196-197 the authors indicate “Among potential protein partners of KaeA, we identified protein kinases (CmkA, PkaA/PkaR, PkcB), putative protein phosphatase, and several proteins involved in translation, such as translation initiation factors, prolyl_tRNA-synthetase and the yeast GCN20 homologue, which was also selected in transcriptomic analysis”.
No, the mRNA for this protein was not selected, I guess the authors want to say this it was observed in transcriptomic studies. Please correct.
3. Finally, although the overall interpretation of the results has been improved the conclusions are still vague and the question remains of what genes are really affected directly (arginine catabolic genes?) or indirectly by the KaeA protein. Are all effects of KaeA exclusively due to the interaction with other proteins of the chromatine complex. Please clarify
Author Response
We thank the Reviewer for comments and suggestions. Response to the Reviewer questions/suggestions are placed below each of the Reviewer's point. (We noticed that line numbering may slightly differ, therefore we give the beginning of the respective fragment of the manuscript)
- As indicated by the reviewer in point 1 of the previous evaluation, the authors postulate, the KaeA bind to some genes, such as the arginine catabolic genes, but they do not clarify this point. Is this based in real experimental results? or it is simply a postulate of the authors that is not supported by the present evidence?. Please clarify
The hypothesis that in addition to its conserved role in tRNA modification, KaeA may be involved in regulation of transcription of arginine catabolism genes was based on observation that their transcription level is high in both kaeA mutants, despite the lack of an arginine inducer in the medium. The results of these experiments were published previously (ref. No 22), and confirmed by the repeated, present transcriptomic analysis. This is described in:
Introduction (lanes 71-75 - "In the model filamentous fungus Aspergillus nidulans,....)
Introduction (lanes 86-90 - " Our previous results supported the hypothesis....),
Results (lanes 140-142 - " As expected, arginine catabolism genes...)
and Discussion (lanes 256-267 - " However, kaeA19 and kaeA25 mutations...).
We did not postulated that KaeA binds to some genes, like arginine catabolic genes. As described in Introduction, we postulated that the expression of some genes, e.g., arginine catabolism genes, might be directly regulated at the transcription level by KaeA (lanes 86-91 - " Our previous results supported the hypothesis....).
In our opinion, the term "direct regulation" referes not only to proteins that bind directly to the DNA of the promoter, but also to co-repressors, co-activators and other co-regulators, which do not bind to DNA, but interact with proteins bound with the promoter region and significantly influence the level of expression of target genes. This is why, according to the Reviewers suggestions, in the revised version of the manuscript we refer to KaeA as " a chromatin - associated protein" and write about "chromatin regions interacting with KaeA", not "genes interacting with KaeA".
In the present manuscript we have shown that, although the (t6A37) tRNA modification is unaffected in both kaeA mutants, they reveal significantly altered transcriptomes compared to the wild type, supporting the hyphothesis that KaeA takes part in the regulation of gene expression. We think that since the promoters of the arginine cataboism genes were not identified in the ChIP-Seq analysis, it is most likely that KaeA regulates these genes indirectly by affecting the expression of one of their regulatory genes. To remove doubts, an appropriate statement was added to the Discussion (lines295-297: "In this way, KaeA can indirectly influence the expression of many more genes regulated by these regulatory proteins, including arginine catabolism genes".)
- In the modified version, lines 196-197 the authors indicate “Among potential protein partners of KaeA, we identified protein kinases (CmkA, PkaA/PkaR, PkcB), putative protein phosphatase, and several proteins involved in translation, such as translation initiation factors, prolyl_tRNA-synthetase and the yeast GCN20 homologue, which was also selected in transcriptomic analysis”.
No, the mRNA for this protein was not selected, I guess the authors want to say this it was observed in transcriptomic studies. Please correct.
This was corrected as follows: "Among potential protein partners of KaeA, we identified protein kinases (CmkA, PkaA/PkaR, PkcB), putative protein phosphatase, and several proteins involved in translation, such as translation initiation factors, prolyl_tRNA-synthetase and the yeast GCN20 homologue, mRNA of which was also identified by transcriptomic analysis" .(lanes 195-198)
- Finally, although the overall interpretation of the results has been improved the conclusions are still vague and the question remains of what genes are really affected directly (arginine catabolic genes?) or indirectly by the KaeA protein. Are all effects of KaeA exclusively due to the interaction with other proteins of the chromatine complex. Please clarify
We hope, that regarding arginine catabolism genes, the answer in point 1 is satisfied.
Yes, we postulate that KaeA effect on transcription is due protein-protein interactions and this is written in conclusions: "As KaeA is a chromatin-associated protein but not a DNA-binding protein, its interactions with chromatin are most probably indirect and mediated via protein–protein interactions, i.e., via proteins that directly bind to DNA or are part of DNA-binding complexes" (lanes 390-392)
We think, that as is written in the Discussion " It seems likely that KaeA, as a chromatin-associated protein, might directly affect the transcription of genes identified in ChIP-Seq analysis." (lanes338-339).
With the term "indirect regulation" we describe the influence of KaeA on gene expression by regulating the expression of their regulatory factors, and this is desribed in:
Results (l. 146-149 - " The observed changes in the expression levels of so many genes ...
and 167-169 - "These results show that KaeA affects the level of transcription of several genes encoding wide-domain regulatory proteins, indirectly influencing the expression of the genes they regulate".
and Discussion (l. 298-316 - " Many genes whose transcription level is altered ....
We also think that at this stage of the research, it would be too speculative to claim that a particular gene is regulated more directly or indirectly by KaeA. This complex regulatory network needs and will be the subject of our further research.
Round 3
Reviewer 1 Report
The authors have reply satisfactorily to the queries of the reviewers and have modified the Results and Discussion accordingly